# A Novel p-Type ZnCo_x_O_y_ Thin Film Grown by Atomic Layer Deposition

**DOI:** 10.3390/nano12193381

**Published:** 2022-09-27

**Authors:** Leyi Li, Zhixin Wan, Quan Wen, Zesheng Lv, Bin Xi

**Affiliations:** 1School of Materials Science and Engineering, Guangzhou Key Laboratory of Flexible Electronic Materials and Wearable Devices, PFCM Lab, Sun Yat-sen University, Guangzhou 510006, China; 2School of Electronics and Information Technology, Sun Yat-sen University, Guangzhou 510006, China

**Keywords:** atomic layer deposition (ALD), ZnCo_x_O_y_ film, p-type metal oxides

## Abstract

Reported herein is the atomic layer deposition (ALD) of novel ternary ZnCo_x_O_y_ films possessing p-type semiconducting behavior. The preparation comprises of optimized ZnO and Co_3_O_4_ deposition in sub-cycles using the commercially available precursors cyclopentadienylcobalt dicarbonyl (CpCo(CO)_2_), diethylzinc (DEZ) and ozone (O_3_). A systematic exploration of the film’s microstructure, crystallinity, optical properties and electrical properties was conducted and revealed an association with Zn/Co stoichiometry. The noteworthy results include the following: (1) by adjusting the sub-cycle of ZnO/ Co_3_O_4_ to 1/10, a spinel structured ZnCo_x_O_y_ film was grown at 150 °C, with it exhibiting a smooth surface, good crystallinity and high purity; (2) the material transmittance and bandgap decreased as the Co element concentration increased; (3) the ZnCo_x_O_y_ film is more stable than its p-type analog Co_3_O_4_ film; and (4) upon p-n diode fabrication, the ZnCo_x_O_y_ film demonstrated good rectification behaviors as well as very low and stable reverse leakage in forward and reverse-biased voltages, respectively. Its application in thin film transistors and flexible or transparent semiconductor devices is highly suggested.

## 1. Introduction

At present, n-type oxide semiconductor thin film materials are widely applied, with their performance being continuously improved. In contrast, p-type oxide semiconductor films are typically poor and remain in the laboratory research stage because their electrical properties are several orders of magnitude lower than that of n-type oxide films, which directly leads to the formation of high potential barriers at the interface between p-type and n-type layers in semiconductor components and results in performance deterioration [1,2]. Thus, the p-type materials, with superior performance, especially those with a high carrier concentration, high hall mobility and low resistivity, still need to be further explored [3] with the aim of achieving mutually compatible p-type and n-type materials, which are in high demand for applications and development in transistor devices, solar cells, sensors and other fields [4].

Spinel ZnM_2_O_4_ (M = Co, Rh or Ir) oxides, with a composite oxide of Zn and group VIII elements, are considered to be a possible intrinsic p-type semiconductor material according to the chemical modulation of the valence band theory carried out by Hiroshi Kawazoe et al. in 1997 [5,6]. Up until now, the p-type conducting behavior of these oxides has been successfully obtained via a series of synthesizing methods, including the liquid phase method (Sol-gel) [7], hydrothermal method [8], solid phase reaction method [9], pulsed laser deposition (PLD) [10] and magnetron sputtering [11], etc. However, the thin films prepared by the above methods are of poor density and have a certain number of defects (i.e., discontinuous growth, holes), which seriously restrict their electrical conductivity [12]. Furthermore, the growth of thin films in various electronic devices of decreasing sizes must be considered, especially when it comes to the stringent growth on complex three-dimensional substrate [13,14]. Therefore, the above techniques can hardly meet the requirements of new semiconductor devices.

As a state-of-the-art thin-film technology, atomic layer deposition (ALD) has a strong applicability in preparing p-type semiconductor films due to its unique characteristics. It is more effective in obtaining highly uniform, conformal and dense film by controlling the thickness and composition under a sequential, self-limited chemisorbed reaction between the precursor and co-reactant [15]. There are a number of successful examples using ALD to prepare p-type conductive films, such as copper-iron series compounds, layered oxygen-chalcogenides and doped binary oxides, etc. [4,6,16]. In spite of these intensive efforts, the tunability of ternary ALD-ZnCo_x_O_y_ metal oxide with varying chemical compositions and structures so far remains rarely reported on. In 2012, M. Godlewski’s group reported a ZnCoO film grown by ALD. However, they were mainly concerned with its ferromagnetic property, but its electrical properties, such as its carrier concentrations, Hall mobility and resistivity, were not revealed [17]. So, a thermal ALD process for growing ZnCo_x_O_y_ at moderate temperatures, specifically a film with good p-type conductive behavior, is therefore in urgent need of being engineered into high-quality thin films to be applied in frontier fields.

Herein, we report the successful growth of cubic spinel ZnCo_x_O_y_ films through a thermal ALD process for the first time by using cyclopentadienylcobalt dicarbonyl (CpCo(CO)_2_), and diethylzinc (DEZ) as Co and Zn precursors, respectively, and ozone (O_3_) as an oxygen source. The comprehensive study of their typical ALD characteristics, i.e., growth characteristics, composition, phase structure evolution as well as the electrical properties of a p-n diode on n-GaN/sapphire substrate, were undertaken to determine the feasibility of them acting as semiconducting materials.

## 2. Experimental Section

### 2.1. Deposition of ZnO, Co_3_O_4_, ZnCo_x_O_y_ Films

*Binary ZnO and Co_3_O_4_ film.* DEZ (99%, Shanghai Oriphant Chemistry Co., Ltd., Shanghai, China), CpCo(CO)_2_ (99%, Shanghai Oriphant Chemistry Co., Ltd., Shanghai, China) and O_3_ were used as Zn, Co precursors and oxidant, respectively. High purity nitrogen (N_2_, 99.999%) was used as a carrier gas and purging gas, with a flow rate of 100 sccm. All depositions were performed in a commercial ALD reactor (PICOSUN^TM^ R-200 Advanced) on substrates of silicon (Si, 100). For CpCo(CO)_2_ supply and delivery, the container of CpCo(CO)_2_ was heated to 50 °C and the DEZ was maintained at room temperature. A unit ZnO cycle consists of: 0.1 s pulse of DEZ, 5 s of purge, 0.1 s pulse of O_3_, 5 s of purge, and that of Co_3_O_4_ involved 0.3 s pulse of CpCo(CO)_2_, 5 s of purge, 5 s pulse of O_3_, 5 s of purge.

*Ternary ZnCo_x_O_y_ film.* To obtain an optimal element ratio to realize p-type conductivity, the ternary ZnCo_x_O_y_ films were grown by varying the sub-cycles of two binary processes with ZnO: Co_3_O_4_ of 1:1, 1:5, 1:10, being denoted ZCO (1:1), ZCO (1:5) and ZCO (1:10). Substrates, including Si, glass and n-GaN on sapphire, were used to facilitate different tests. The detailed optimization of the deposition conditions and the variations in all of the ALD parameters are described in the Results and Discussion section.

### 2.2. Characterizations of ZnO, Co_3_O_4_, ZnCo_x_O_y_ Films

For the as-deposited binary and ternary thin film, the thickness was measured by ellipsometry (J. A. Woollam Co., Guangzhou, China) alpha-SE). The surface and cross-sectional morphology were studied by atomic force microscope (AFM, Bruker-icon Guangzhou, China). Crystal structural information was collected from X-ray powder diffraction (XRD) in the 2θ range of 10–80° (GIXRD, smartlab 9 kw, Rigaku, Wuhan, China), and the electrical properties were measured by a Hall measurement (JouleYacht HET-RT, Guangzhou, China). The film composition, chemical bonding and bandgap were analyzed by X-ray photoelectron spectroscopy (XPS, Thermo ESCALAB 250XI, Wuhan, China) upon 30 s surface etching (2 kV Ar^+^, to remove surface oxides and carbon). Optical transmission was also tested by ultraviolet-visible spectrophotometer (Shimadzu UV2550, Guangzhou, China).

### 2.3. Electrical Property of Diode Fabricated by ZnCo_x_O_y_ Film

A p-n diode was fabricated on n-GaN/sapphire substrate by using GaN as the n-layer and ZnCo_x_O_y_ as the p-layer, in which the n-GaN layer with a thickness of 3.5 μm was prepared by a CVD process at 1050 °C and the ZnCo_x_O_y_ p-layer with a thickness of approximately 30 nm was constructed under 150 °C. On the surface of the n- and p-type layers, a mesa etching was firstly executed by an inductively coupled plasma process down to the n-GaN layer using a BCl_3_/Cl_2_ gas source. Then, a Ti/Al/Ni/Au multilayer and Pt metal were deposited by electron-beam evaporation and annealed by a rapid thermal annealing process at 600 °C for 2 min in N_2_ ambient to form the ohmic contacts. The *I-V* characteristics of the diode were measured using a Keithley 4200-SCS semiconductor characterization system.

## 3. Results and Discussion

*Growth of oxide thin films.* The saturation behaviors of both ZnO and Co_3_O_4_ were first studied at 150 °C by varying the pulse length of one precursor, with the others being fixed. As shown in Figure 1a,d, the DEZ and CpCo(CO)_2_ pulse length for the saturation growth of ZnO and Co_3_O_4_ are 0.3 s and 0.5 s, respectively. Regarding the experiment of O_3_ pulse change, the saturation of growth of ZnO begins at a pulse length of 0.1 s and that of Co_3_O_4_ requires a longer pulse of 5 s (Figure 1b,e), which leads to a similar growth rate of 0.06 nm/cycle for both oxide thin films. Such observed saturation behavior confirms a typical ALD process for the binary films (Co_3_O_4_ and ZnO) by using DEZ, CpCo(CO)_2_ and O_3_. In order to determine a suitable growth temperature for ZnO film, the depositions were performed in a temperature ranging from 75–300 °C by pulsing precursors of DEZ and O_3_ alternately at lengths of 0.3 s and 0.1 s, respectively. As shown in Figure 1c, the growth rate of ZnO film, which is approximately 0.06 nm/cycle, remains stable from 150–250 °C, indicating a wide ALD window. As for the Co_3_O_4_ films, the growth rate slightly increases from 0.055 to 0.06 nm/cycle with an increase in temperature from 100 °C to 175 °C and the maximum growth rate (0.065 nm/cycle) is observed at 200 °C, with the rate then decreasing with further increases in temperature. Thus, a relative narrow ALD window, ranging from 125 °C to 175 °C, is estimated for Co_3_O_4_ growth.

When it comes to the growth temperature of the ternary oxide, it should be in the ALD windows of both binary oxides concurrently; therefore, the deposition temperature for ZnCo_x_O_y_ growth should be performed at 150 °C. The optimal process parameters, obtained from the above ALD characteristic curves, were used for growth of the ZnCo_x_O_y_ composite films, that is, a 0.3 s pulse of Zn precursor and a 0.1 s pulse of O_3_ for ZnO growth, a 0.5 s pulse of Co precursor and a 5 s pulse of O_3_ for Co_3_O_4_ growth, with 5 s N_2_ purging.

*Characterization of**oxide thin films.* The morphologies of the as-deposited pure ZnO, Co_3_O_4_ and ZnCo_x_O_y_ were detected by AFM and are displayed in Figure 2. A homogeneous surface with small particles can be clearly observed for all samples. In the case of the ZnCo_x_O_y_ films, the surface tends to agglomerate, with nano-grains approaching each other, resulting in a tightly packed morphology as the Co_3_O_4_ sub-cycles ratio increases. All the ternary films have a smooth surface and a low root mean square roughness (Rq, about 2 nm), which is in accordance with the fact that the hybrid deposition process is favorable for ZnCo_x_O_y_ crystal growth.

Figure 3 shows the XRD patterns of pure ZnO, Co_3_O_4_ and ZnCo_x_O_y_ films for various binary oxide ratios. Both binary oxides have good crystallinity. The single-phase ZnO, with peaks located at 31.7°, 34.4°, 36.3°, 47.5° and 62.9°, can be assigned to the (110), (002), (101), (102) and (103) planes of a hexagonal structure (JCPDS file No. 05-0664). For the pure Co_3_O_4_ film, all reflection peaks, 31.2°, 36.8°, 44.8° and 59.4°, are indexed to the (220), (311), (400) and (511) planes of a cubic spinel structure (JCPDS file No. 09-0418). In the case of the ternary ZnCo_x_O_y_ films, a structural evolution as a result of different elemental ratios can be clearly seen. ZCO (1:1) shows similar diffraction peaks with the same position of pure ZnO, and only the (110) and (101) planes are mildly inhibited, which reveals that ZnO is the main composite in the film and that the Co_3_O_4_ sub-cycle grown on the ZnO surface may be slightly suppressed during the deposition process. The ZnCo_x_O_y_ exhibits an amorphous structure, with the Co_3_O_4_ sub-cycle ratio increasing to 5. As expected, a spinel ZnCo_x_O_y_ film with good crystallinity is achieved under a 1:10 deposition ratio of ZnO:Co_3_O_4_. Contrasting with the crystal structure of ZnCo_2_O_4_ (JCPDS file No. 23-1390), five diffraction peaks are located at 2θ = 30.9°, 36.3°, 44.1°, 58.4° and 55.6° and can be assigned to the (220), (311), (400), (511) and (440) planes of the spinel structure. No additional secondary phase and impurity phases related to Zn/Co were detected, suggesting that the Co^2+^ ions are partly occupied by the Zn^2+^ in the ZCO structure. Moreover, a slight shift a to lower peak position can be observed from the XRD curves, which we suspect indicates that a small amount of Zn^2+^ may replace the Co^3+^ ions sites (doping effect) or that partial Zn^2+^ dissolves into the ZnCo_x_O_y_ as interstitial atoms, giving rise to the change in terms of the stoichiometric composition and lattice deformation of ZnCo_x_O_y_ films, causing further peak shifting.

Considering that the goal of this research was to obtain p-type films with a spinel structure, the ZnCo_x_O_y_ (ZCO 1:10) was used as a typical sample for XPS measurements and its chemical composition (Appendix A, “S” denotes Appendix A), element oxidation state and the existence of an oxygen vacancy was analyzed in detail. The high resolution XPS spectra of Zn 2p, Co 2p and O 1 s are shown in Figure 4. The Zn 2p spectrum, as presented in Figure 4a, has two main peaks, Zn 2p1/2 and Zn 2p3/2 (1020.3 and 1043.5 eV), which belong to the 2+ oxidation state of Zn [18,19]. As for the Co 2p spectra in Figure 4b, the peaks at 779.3 and 794.5 eV can be reasonably assigned to Co 2p3/2 and Co 2p1/2, with spin-orbit splitting of around 15 eV [20], in which, the presence of both divalent and trivalent Co cation can be deconvoluted by the binding energies at 779.4 eV and 780.5 eV for Co^3+^ and Co^2+^ (2p3/2), respectively. Additionally, the O 1s XPS spectrum in Figure 4c presents a lower energy peak at 529.4 eV and an affiliated peak of 531.2 eV, with this being attributed to metal–oxygen bonding (oxygen bonding with Zn and Co atoms) and a small amount of oxygen vacancies (O_v_) in the ZnCo_x_O_y_ film [20,21]. The poor O_v_ density in the ZnCo_x_O_y_ (ZCO 1:10) film benefits its conductivity because the oxygen in the p-type material can act as an acceptor and the substantially suppressed O_v_ allows enough oxygen to generate free holes to enhance its p-type conductivity [22]. The above XPS analysis verifies that the valences of Zn, Co and O elements are +2, +2/+3 and −2, respectively, which is consistent with the element chemical states in ZnCo_2_O_4_ compound.

*Optical properties of oxide thin films.* Figure 5a shows the optical transmittance spectra of the pure ZnO, Co_3_O_4_ and ZnCo_x_O_y_ films. All films exhibit a transmittance band in the visible region and an absorption band edge in the ultraviolet. As it can be observed, pure ZnO film exhibits an excellent transmittance of about 90%, indicating a good transmittance performance as a mature transparent oxide conducting material. As for the pure Co_3_O_4_ film, a poor transmittance of 30% can be detected, and the obtained spectrum presents two absorption signals in the visible region. The first one (300 nm ~ 400 nm) is associated to the O^2–^ and Co^2+^ charge transfer process, and the other (500 nm~ 600 nm) is from the O^2–^ and Co^3+^ charge transfer, with this being in agreement with the literature [23]. Meanwhile, the ZCO (1:1) films exhibit a remarkable transmittance close to that of pure ZnO, which confirms that the ZCO film with a 1:1 deposition ratio is dominated by ZnO composite while the growth of Co_3_O_4_ is inhibited, with this phenomenon being consistent with the XRD results. Additionally, it is evident that the transmittance gradually decreases as the number of Co_3_O_4_ sub-cycle increases, emphasizing the enhancing of the proportion of Co_3_O_4_ in the ratio. From the (αhυ)^2^ versus hυ plots shown in Figure 5b, the band gap can be found to decrease with the increase in Co content. The bandgap of ZCO (1:1) is estimated to be 3.2 eV, with this being close to the value of pure ZnO (3.3 eV), and that of ZCO (1:10) is approximately 2.15 eV, showing a slightly higher value than 1.85 eV of pure Co_3_O_4_. As for ZCO (1:5), it is difficult to determine its bandgap because no obvious absorption edge is presented from the curve related to the optical band gap of a semiconductor.

*Electrical properties of oxide thin films.* Room temperature Hall measurements were carried out to determine the conductivity-types, carrier concentrations and mobilities of the thin films, as listed in Appendix A. The pure Co_3_O_4_, ZCO (1:5) and ZCO (1:10) films exhibit typical p-type conductivity behavior, among which ZCO (1:10) shows the best properties with a resistivity of 0.23 Ω·cm, a carrier concentration of 2.46 × 10^19^ cm^−3^ and a mobility of 1.09 cm^2^V^−1^s^−1^, suggesting the successful design of spinel p-type ZnCo_x_O_y_ films in this work.

By taking advantage of the p-type conductivity in ALD-grown ZnCo_x_O_y_ thin films, a p-n diode on n-GaN/sapphire substrate was fabricated. Figure 6a shows the schematic diagram of the completed diode built by a general model as described in the Experimental section. The ohmic contacts and I-V characteristics for the p-n junctions are depicted in Appendix A and Figure 6b. To validate the p-n junctions, *I-V* characteristics between the p-type layer and a platinum electrode of the diodes were measured (inset in Figure 6b). In all cases, symmetric *I-V* characteristics can be observed to ensure that the rectification behavior of the diode arises from the p-n junction. As a result, the three junctions showed good rectification behaviors in a voltage range of ±5 V. Under forward-biased voltage, we obtained an ideality factor of n = 1.5 and 2.4 V of forward turn-on voltage from the p-ZCO (1:10)/n-GaN diodes (Appendix A). Moreover, the rectification of p-ZCO (1:10)/n-GaN diodes is 1000 times higher than that of a pure Co_3_O_4_ based p-n junction. In a reverse-biased voltage range, a very low and stable reverse leakage current of about 2 fA was obtained for the p-ZCO (1:10)/n-GaN diode, corresponding to a current density of 0.11 nA/cm^2^. Meanwhile, the ratio of forward-to-reverse current (I_f_/I_r_) was 1.94 × 10^9^ when ZCO (1:10) film was used as the p-layer, which is three orders of magnitude higher than that of a pure Co_3_O_4_ film-based p-layer (6.53 × 10^6^). The remarkable performance of ZCO (1:10) film may be due to its ideal spinel structure and possible antisite defects of Zn_Co_ in the lattice [24]. The p-ZCO (1:5)/n-GaN diode also exhibited a low leakage current similar to p-ZCO (1:10)/n-GaN under −3 V reverse-biased voltage, but the leakage current significantly increased with an increase in the reverse-biased voltage, indicating a poor stability when compared with that of p-ZCO (1:10)/n-GaN, probably due to its amorphous structure. The aforementioned XRD result (Figure 3) shows that the crystal structure of ZCO (1:5) was totally disrupted and that a large amount of internal distortion may generate inside the film, thus leading to a negative effect on its performance. No curve for the ZCO (1:1)-based diode was presented here since it exhibited n-type conductivity behavior, indicating that the film with a 1:1 sub-cycle ratio still remains an intrinsic property of ZnO. As mentioned above, the growth of the Co_3_O_4_ sub-cycle grown on the ZnO surface is suppressed, and thus leads to a low level of Co element in the ZCO (1:1) film. Based on our analysis, it is considered that a very small amount of Co^2+^ may act as a doping element to replace Zn^2+^, but its resistivity and conductivity as well as the crystal structure would not be significantly affected.

## 4. Conclusions

This work described the growth of a p-type spinel ZnCo_x_O_y_ film utilizing a thermal ALD technique via repeated cycles combining binary ZnO and Co_3_O_4_ sub-cycles. Phase evolution could be observed by adjusting the sub-cycle ratio, and sufficiently crystallized ZnCo_x_O_y_ films can be obtained under an optimal ratio of ZnO:Co_3_O_4_ = 1:10. The as-deposited ZnCo_x_O_y_ are relatively uniform and continuous, with a roughness of about 2 nm. High-purity film with a small amount of O_v_ may be of benefit and result in a resistivity of 0.23 Ω·cm, carrier concentration of 2.46 × 10^19^ cm^−3^ and mobility of 1.09 cm^2^V^−1^s^−1^. The ZnCo_x_O_y_ (1:10) film-based p-n junction presents good rectification behaviors in a voltage range of ±5 V and low leakage current with high stability under reverse-biased voltage. We anticipate that this work will pave the way for the development of a series of p-type multiple composites with excellent properties via the ALD process, which plays a crucial role to contributing to the development of new semiconductor materials.

## Figures and Tables

**Figure 1 nanomaterials-12-03381-f001:**
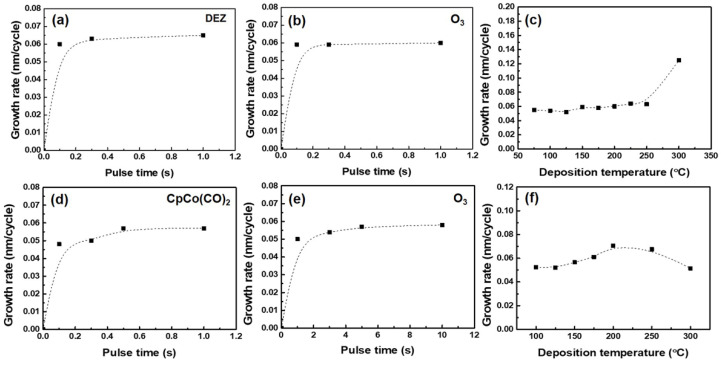
The ALD saturation curves and deposition windows of (**a**–**c**) pure ZnO and (**d**–**f**) Co_3_O_4_ deposited by DEZ, CpCo(CO)_2_ and O_3_, respectively, with 300 deposition cycles on Si.

**Figure 2 nanomaterials-12-03381-f002:**
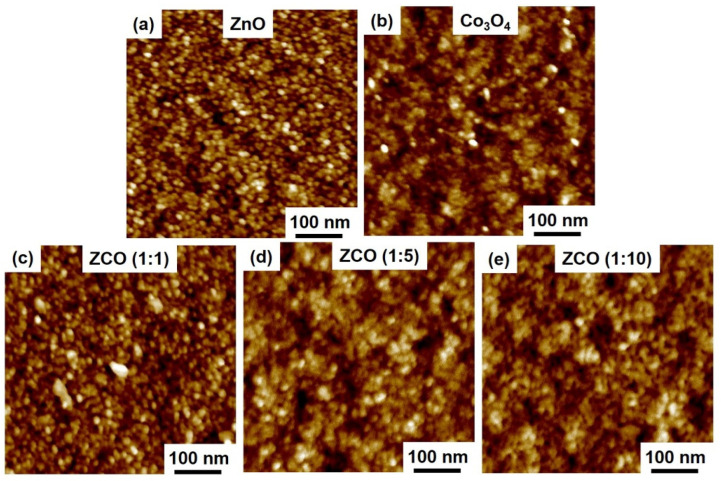
AFM images of the (**a**) pure ZnO, (**b**) Co_3_O_4_ and (**c**–**e**) ZnCo_x_O_y_ films with 1:1, 1:5, 1:10 sub-cycle deposition ratios after 500 cycles on Si.

**Figure 3 nanomaterials-12-03381-f003:**
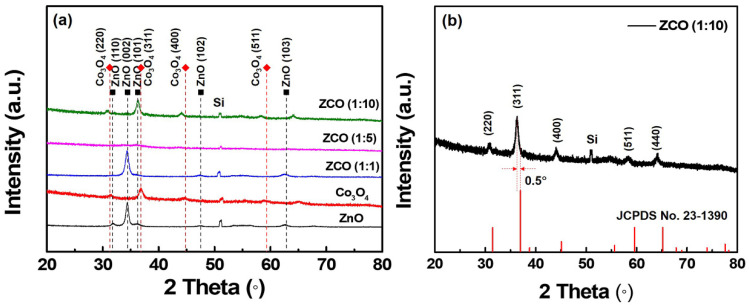
XRD analysis of a (**a**) comparison of the pure ZnO, Co_3_O_4_ and ZnCoxOy films with different deposition ratios and the (**b**) ZnCo_x_O_y_ films (1:10) contrasted with standard cards.

**Figure 4 nanomaterials-12-03381-f004:**
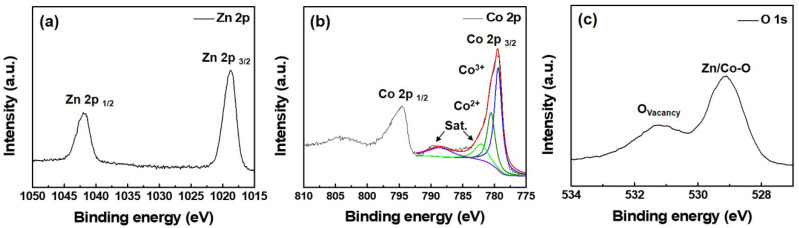
XPS analysis of the (**a**) Co 2p spectrum, (**b**) Zn 2p spectrum and (**c**) O 1s spectrum of the ZnCo_x_O_y_ (ZCO 1:10) film.

**Figure 5 nanomaterials-12-03381-f005:**
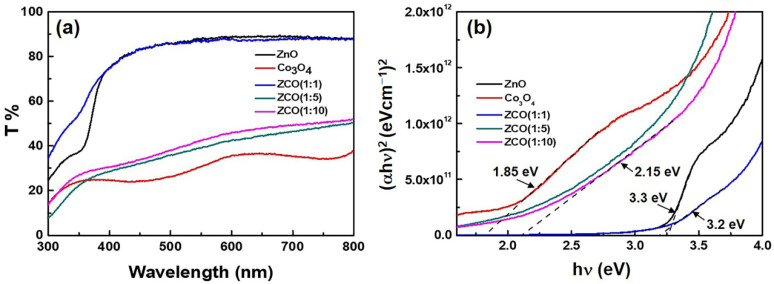
(**a**) Optical transmission and (**b**) Tauc plots of the pure ZnO, Co_3_O_4_ and ZnCo_x_O_y_ films with different deposition ratios.

**Figure 6 nanomaterials-12-03381-f006:**
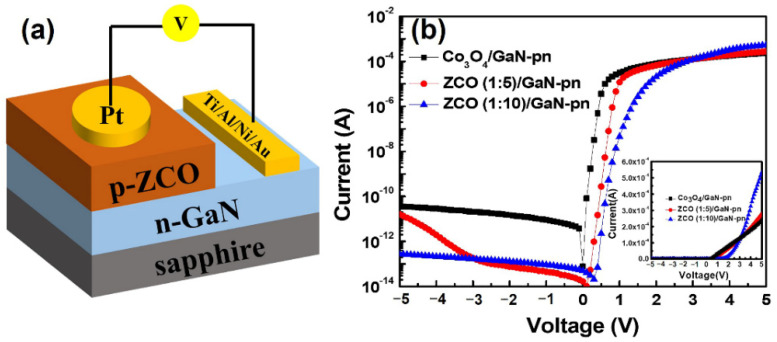
(**a**) schematic diagram of the fabricated p-n diode and (**b**) the corresponding *I-V* characteristics of p-Co_3_O_4_/n-GaN and p- ZnCo_x_O_y_/n-GaN.

## Data Availability

Not applicable.

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
