# Peer review of "A Novel p-Type ZnCoxOy Thin Film Grown by Atomic Layer Deposition"

_nanomaterials, 2022, doi:10.3390/nano12193381_

Round 1

Author Response

Dear reviewer,

Thank you very much for your valuable comments on our manuscript “A Novel p-type ZnCoxOy Thin Film Grown by Atomic Layer Deposition” (nanomaterials-1882777).

We checked your comments carefully and put efforts to revise/improve this work accordingly. Uploaded are the revised manuscript addressing all issues raised, with all changes marked in red for the convenience of your examination. 

We sincerely appreciate the constructive comments from you and hope that you find the revisions satisfactory. 

Thank you very much.

Yours sincerely,

Zhixin Wan

Reviewer 2 Report

The manuscript looks interesting for the development of p-type conducting materials. Here I have following comments, and the authors need to revise the manuscript based on these comments to make the manuscript better. The manuscript can be considered for publication after the authors revise the manuscript based on the following comments.

Comments:

1.       Did the authors carry any elemental analysis for the materials they have grown, for example, EDX? Based on the XRD pattern, it seems like Co incorporates into ZnO but in what ratio for different samples (?). The elemental analysis would be helpful.

2.       Regarding the transmission spectra presented on Figure 5. Why is the T% so low for these ZCO materials relative to ZnO? They have calculated the band gaps for ZCO (1:1) and ZCO (1:10) but not for ZCO (1:5). Do the authors get the same composition and properties of the films fabricated for the same recipe for instance in the case of 1:10? Or it keeps changing with each batch. Also, can the authors show the T% curves for longer wavelength to about 1200 nm or 1500 nm. Do they have low transmission at NIR region as well?

3.       In Hall measurements, why is the carries concentration low (1017 cm-3) for ZCO (1:5) compared to ZCO (1:1) and ZCO (1:10)? What is statistical variation on these measurements on the data shown in Table S1?

4.       The authors claim ZCO (1:10) has the best properties as suggested by the Hall measurements (P9, line 204). What happens if the ratio changes to 1:20 or 1:100? Do they expect even better-quality materials?

5.       In Figure S1, I could not see the curve for ZCO (1:5). Does it overlap completely with other curves? Please, modify appropriately with symbol and markers so it becomes visible to readers.

6.       Will the authors include the I-V curves (linear) for all the cases in Figure 6? The inset looks good for ZCO (1:10). Also, I would encourage to have more discussion regarding the doping (and defects) that come with p-type doping on these materials in the manuscript.

Author Response

(The authors gave the same response as above.)

Round 2

Reviewer 1 Report

The y-axis in Figure 5b needs a scale and units.. 

Author Response

Dear reviewer,

We sincerely appreciate your constructive comments and the mentioned figure was carefully revised as shown in the manuscript.

Thank you very much.

Yours sincerely,

Zhixin Wan

Reviewer 2 Report

The revised manuscript looks good, and recommend accepting the manuscript in its present form.

Author Response

Dear reviewer,

Thank you for your recognition of our revised manuscript.

Yours sincerely,

Zhixin Wan